# Comparative Efficacy of ALK Inhibitors for Treatment-Naïve ALK-Positive Advanced Non-Small Cell Lung Cancer with Central Nervous System Metastasis: A Network Meta-Analysis

**DOI:** 10.3390/ijms24032242

**Published:** 2023-01-23

**Authors:** Koichi Ando, Ryo Manabe, Yasunari Kishino, Sojiro Kusumoto, Toshimitsu Yamaoka, Akihiko Tanaka, Tohru Ohmori, Hironori Sagara

**Affiliations:** 1Division of Respirology and Allergology, Department of Medicine, Showa University School of Medicine, 1-5-8 Hatanodai, Shinagawa-ku, Tokyo 142-8666, Japan; 2Division of Internal Medicine, Showa University Dental Hospital Medical Clinic, Senzoku Campus, Showa University, 2-1-1 Kita-senzoku, Ohta-ku, Tokyo 145-8515, Japan; 3Advanced Cancer Translational Research Institute, Hatanodai Campus, Showa University, 1-5-8 Hatanodai, Shinagawa-ku, Tokyo 142-8555, Japan; 4Department of Medicine, Division of Respiratory Medicine, Tokyo Metropolitan Health and Hospitals Corporation, Ebara Hospital, 4-5-10 Higashiyukigaya, Ohta-ku, Tokyo 145-0065, Japan

**Keywords:** brain metastasis, ALK rearrangement, acquired resistance, network meta-analysis

## Abstract

Central nervous system (CNS) metastases and acquired resistance complicate the treatment of anaplastic lymphoma kinase (ALK) rearrangement-positive (ALK-p) advanced non-small cell lung cancer (NSCLC). Thus, this review aimed to provide a comprehensive overview of brain metastasis, acquired resistance, and prospects for overcoming these challenges. A network meta-analysis of relevant phase III randomized controlled trials was performed to compare the efficacies of multiple ALK inhibitors by drug and generation in overall patients with ALK-p untreated advanced NSCLC and a subgroup of patients with CNS metastases. The primary endpoint was progression-free survival (PFS). Generation-specific comparison results showed that third-generation ALK inhibitors were significantly more effective than second-generation ALK inhibitors in prolonging the PFS of the subgroup of patients with CNS metastases. Drug-specific comparison results demonstrated that lorlatinib was the most effective in prolonging PFS, followed by brigatinib, alectinib, ensartinib, ceritinib, crizotinib, and chemotherapy. While lorlatinib was superior to brigatinib for PFS in the overall patient population, no significant difference between the two was found in the subgroup of patients with CNS metastases. These results can serve as a foundation for basic, clinical, and translational research and guide clinical oncologists in developing individualized treatment strategies for patients with ALK-p, ALK inhibitor-naive advanced NSCLC.

## 1. Introduction

### 1.1. Overview

The tremendous advances in tumor molecular research over the past two decades have contributed enormously to our understanding of the etiology of non-small cell lung cancer (NSCLC), which constitutes 84% of all primary lung cancers [1,2,3,4,5]. As a result, therapeutic strategies for NSCLC have evolved remarkably and are still undergoing further development [1,3]. Despite these advances, lung carcinoma currently remains the primary cause of cancer-related mortality, representing 13% of all deaths related to cancer. Localized NSCLC has a reported 5-year survival rate of 63%. However, most NSCLC cases are diagnosed in an advanced stage. The 5-year survival rate of patients with progressive cancer is currently approximately 7%, even with current recommended therapeutic regimens [2,5,6]. Thus, further improvements and developments are needed to treat NSCLC. Along with the expansion of cancer genome medicine, genetic diagnostic technologies, treatment options, cancer gene panel tests (cancer genome profiling tests) and liquid biopsy, which examines numerous genes in cancer tissue samples, is being standardized [3,7,8,9,10,11]. These technologies have contributed to a further paradigm shift toward personalized medicine for NSCLC. Three to five percent of NSCLC cases harbor anaplastic lymphoma kinase (ALK) gene fusions, the most common of which is echinoderm microtubule-associated protein-like 4-ALK (EML4-ALK) [12,13,14,15,16].

As shown in Figure 1, EML4-ALK activates downstream signals such as phosphatidylinositol 3-kinase-AKT (PI3K-AKT), Janus kinase/signaling transcription and activation factor (JAK/STAT) signaling cascade, reticular activation system (RAS), and various kinase activity, which consequently promote the production of fusion proteins that inhibit apoptosis [12,17,18,19]. This phenomenon promotes tumor survival, growth, progression, and metastasis to other organs, including metastasis to the central nervous system (CNS), and in different cancer types, including NSCLC [1,2,8,15,18,19,20,21,22,23].

Lung cancer has a high probability of metastasis to the CNS, and patients with advanced NSCLC involving driver gene alterations, such as EGFR mutations and ALK rearrangements, frequently have brain metastases (BM). The incidence of BM in such patients is 37–64% [14,24,25]. Hence, treatment strategies focusing on BM are essential to manage patients with advanced NSCLC, especially those with ALK rearrangements. Whole-brain radiotherapy (WBRT) is the primary treatment option for patients with BM from NSCLC. However, tyrosine kinase inhibitor (TKI)-targeted driver gene alteration has marked antitumor activity against BM in tumors with driver alterations [14,26]. Therefore, systemic pharmacotherapy with TKIs is a promising treatment option for BM from ALK rearrangement-positive (ALK-p) NSCLC. Multiple therapies have been introduced. In 2011, the U.S. FDA first approved crizotinib as the leading ALK inhibitor. Subsequently, second- and third-generation ALK inhibitors have been approved and reported to have superior efficacy and antitumor activity to crizotinib in patients with CNS metastases.

Nevertheless, therapeutic strategies targeting ALK-p NSCLC with CNS metastases remain to be developed, and current strategies need to be improved [2,12,14].

Even patients who respond well to initial therapy with ALK inhibitors may experience tumor recurrence because of acquired resistance to these drugs via multiple mechanisms, including secondary ALK resistance mutations and tumor growth promoting molecular pathways. Therefore, the mechanisms underlying acquired resistance to ALK inhibitors must be elucidated to develop novel strategies to treat patients with ALK-p advanced NSCLC, especially those with CNS metastases [4,12].

In this meta-analysis report we first summarize the different ALK inhibitors (crizotinib, ceritinib, alectinib, brigatinib, antacatinib, and loratinib) used in patients with ALK-p advanced NSCLC. We then discussed the treatment of patients with CNS metastases and the prospects for overcoming acquired resistance mutations in these patients.

### 1.2. ALK Inhibitors

ALK inhibitors bind to the ATP-binding pocket of the intracellular tyrosine kinase domain, and regulate their downstream signals such as the RAS, PI3K-AKT, JAK/STAT signaling cascades which are involved in tumor progression; attenuation of these cascades produces an antitumor effect [8,20,27]. At present, several ALK inhibitors, including crizotinib, alectinib, ceritinib, brigatinib, ensartinib, and lorlatinib, have been approved as standard therapy for ALK-p NSCLC [1,3,17].

#### 1.2.1. Crizotinib

Crizotinib is a first generation ALK inhibitor authorized for the potential therapeutic application in the treatment of ALK-p NSCLC. Two phase III trials (PROFILE1014 [28] and PROFILE1029 [29]) reported that progression-free survival (PFS) is longer in patients treated with crizotinib monotherapy than in those treated with platinum combination therapy (hazard ratio [HR] 0.45, 95% confidence interval [CI]: 0.35–0.60; HR 0.402, 95% CI: 0.286–0.565). Drug resistance usually develops within a year, and the disease metastasizes to the brain because crizotinib cannot penetrate the blood–brain barrier (BBB) [2,30,31,32]. The major toxicities of crizotinib monotherapy include liver dysfunction, visual disturbances, and gastrointestinal toxicities such as diarrhea and nausea. Although PFS is longer with crizotinib monotherapy than with platinum combination therapy, multiple phase III trials have shown that other ALK-TKI monotherapies (alectinib, brigatinib, and lorlatinib) are more effective than crizotinib in prolonging PFS [1,33,34,35]. Previous clinical trials and our previous NMA showed that a toxicity level of Grade 3 or higher is more frequent with crizotinib monotherapy than with alectinib monotherapy [35].

#### 1.2.2. Ceritinib

Ceritinib is a second-generation ALK inhibitor developed to improve the low activity of first-generation ALK-TKIs against CNS diseases and overcome resistance. A phase III study (ASCEND-4) found that the PFS of patients with stage IV ALK-p NSCLC and PS 0–1 is longer with ceritinib monotherapy than with platinum combination therapy (HR 0.55, 95% CI: 0.42–0.73) [36]. However, Grade 3 or higher adverse events are more frequent with ceritinib monotherapy (65%) than with platinum combination therapy (49%). The main toxicities of ceritinib include liver dysfunction and gastrointestinal toxicities, including anorexia, diarrhea, nausea, and vomiting [1,36].

#### 1.2.3. Alectinib

Three phase III trials (J-ALEX [37], ALEX [35], and ALESIA [38]) involving patients with ALK-p stage IV NSCLC and PS 0–1 found that PFS is significantly longer in patients treated with alectinib monotherapy than in those treated with crizotinib monotherapy (HR 0.38, 95% CI: 0.26–0.55; HR 0.47, 95% CI: 0.34–0.65; HR 0.22, 95% CI: 0.13–0.38). Updated results of the ALEX trial reported that overall survival (OS) is longer with alectinib monotherapy than with crizotinib monotherapy (HR 0.67, 95% CI: 0.46–0.98) [35]. The J-ALEX study showed that like ceritinib, alectinib has a high response rate and excellent brain penetration and that Grade 3 or higher adverse events are less frequent with alectinib monotherapy (32%) than with crizotinib monotherapy (57%) [37]. Our previous NMA has shown that alectinib monotherapy has superior efficacy to other ALK inhibitors [39]. The main toxicities of alectinib monotherapy include dysgeusia, myalgia, and skin rash; as with other kinase inhibitors, interstitial pneumonitis should also be noted [1,35,37,38].

#### 1.2.4. Brigatinib

A phase III trial (ALTA-1L [34]) that included patients with stage IV ALK-p NSCLC and PS 0–1 revealed that the PFS of these patients is longer with brigatinib monotherapy than with crizotinib monotherapy (HR 0.49, 95% CI: 0.33–0.74) [34]. Our NMA reported that brigatinib monotherapy has a relatively favorable efficacy in patients with BM. However, no significant difference was demonstrated when brigatinib monotherapy was compared with alectinib monotherapy [39,40]. Grade 3 or higher adverse events are more frequent with brigatinib monotherapy (61%) than with crizotinib monotherapy (55%) [34]. The major toxicities associated with brigatinib monotherapy include hypertension, interstitial pneumonia, elevated creatine kinase, skin rash, and gastrointestinal toxicities, such as nausea, vomiting, and diarrhea [1,34].

#### 1.2.5. Ensartinib

A phase III eXalt3 trial demonstrated that ensartinib (X-396) monotherapy, as a first-line treatment for patients with ALK-p advanced NSCLC, is better than crizotinib monotherapy in prolonging PFS (HR 0.51, 95% CI: 0.35–0.72) [41]. In addition, the intracranial response efficiency of ensartinib monotherapy is higher (63.6%; *n* = 7/11 patients) than that of crizotinib monotherapy (21.1%; *n* = 4/19 patients). However, the incidence of treatment-related serious adverse events is higher with ensartinib monotherapy (7.7%) than with crizotinib monotherapy (6.1%), with no new safety signals [41].

#### 1.2.6. Lorlatinib

A phase III trial (CROWN [33]) showed that the PFS of patients with stage IV ALK-p NSCLC and PS 0–1 is significantly longer with lorlatinib monotherapy than with crizotinib in the overall patient population and a subgroup of patients with CNS metastases (HR 0.28, 95% CI: 0.19–0.41). However, the incidence of Grade 3 or higher adverse events is higher (72%) with lorlatinib monotherapy than with crizotinib monotherapy (56%). The major toxicities of lorlatinib monotherapy include hypercholesterolemia, hypertriglyceridemia, weight gain, and hypertension, with cognitive dysfunction (2%) reported as the most common adverse event [1,33].

### 1.3. Current Insights and Future Prospects on Treatment Strategies for ALK-p NSCLC with BM

In this section we discuss the treatment options for asymptomatic and symptomatic BM cases and the future direction of drug therapy development.

For asymptomatic BM cases, single-drug therapy with TKIs is the preferred and recommended treatment option because the patients’ tumors, including BM, are expected to have high responsiveness to these drugs [14,26].

However, BM progression can easily cause neurological symptoms and rapidly deteriorate the patient’s condition. For example, BM lesions in the brain stem or close to the pyramidal tract can rapidly deteriorate, even if their size is small. Thus, intracranial radiotherapy is preferred for such patients, even those who are asymptomatic. Close monitoring of BM and the timing of radiotherapy intervention are critical in managing asymptomatic cases [1,12,14]. 

For symptomatic BM cases intracranial radiotherapy is the primary treatment option because robust local control of BM and neurological improvement are expected. The drugs selected by treating physicians are also essential in managing symptomatic BM cases. Several phase III clinical trials demonstrated that the intracranial antitumor activity of second- or further-generation ALK-TKIs is higher than that of crizotinib [42,43]. Lorlatinib, a third-generation ALK-TKI that can be delivered to the CNS, shows robust intracranial tumor response [33]. In a previous clinical trial the objective response rate (ORR) of lorlatinib (82%) in ALK-TKI-naïve patients with BM is higher than that of crizotinib [44]. However, hyperlipidemia and neurological adverse events such as cognitive impairment, anxiety, and depression are specific to lorlatinib [44]. Given the differences in adverse events induced by ALK-TKIs [33,42,43,45], patients’ conditions and concomitant diseases should be considered when selecting ALK-TKIs. 

Drug delivery into the CNS is generally prevented and primarily regulated by the BBB. WBRT can irreversibly disrupt the BBB and improve the delivery of ALK-TKIs [46,47]. WBRT is effective against BM and can enhance drug delivery to intracranial lesions. However, the incidence of cognitive impairment after WBRT is higher than that after stereotactic radiotherapy [48]. Assumption of the remaining neurological function after cranial radiotherapies and individualized treatment options should be discussed on a case-by-case basis by a multidisciplinary team.

Drug therapy combined with other agents and fourth-generation ALK-TKIs are future treatment options for ALK-p NSCLC with BM.

Glycogen synthase kinase 3 (GSK3) is a serine-threonine kinase that serves an essential role in many cellular processes; thus, it has been considered as a promising target in various malignant diseases including brain tumors [49]. A preclinical study demonstrated that a GSK3 inhibitor in combination with lorlatinib can overcome lorlatinib resistance [50]. In addition to its antitumor effects, GSK3 inhibition also exerts neuroprotective effects by promoting DNA repair. Thus, lorlatinib treatment combined with GSK3 inhibitors can be more potent and neurologically less toxic for patients with BM from ALK-p NSCLC. Fourth-generation ALK-TKIs, such as TPX-0131 and NUV-655, have been developed and are now under investigation. These drugs can penetrate the CNS and thus are more potent than conventional ALK-TKIs [51,52]. They are active against ALK-p NSCLC with L1196M and G1202R mutations, which are resistant to third-generation ALK-TKIs such as lorlatinib [51,52]. Thus, fourth-generation ALK-TKIs are a promising treatment option for BM from ALK-p NSCLC, especially for those cases where other ALK-TKIs failed.

Radiotherapy, drug therapy, and their combinations have markedly progressed and prolonged the survival of patients with BM from ALK-p NSCLC. However, individualized treatment selection is a critical component for managing such patients. Thus, a multidisciplinary team comprising thoracic and radiation oncologists should conduct individualized treatment discussions to prolong the survival of patients with BM from ALK-p NSCLC.

### 1.4. Mechanisms of Acquired Resistance to ALK Inhibitors and Prospects for Novel Strategies

Although crizotinib can significantly prolong the response rate and PFS of patients with ALK-p NSCLC, disease progression inevitably occurs after treatment because of the acquired resistance of 1–2 years [33]. The mechanisms underlying the acquired resistance to ALK inhibitors are classified into two categories: (1) ALK-dependent resistance mechanisms, such as secondary ALK resistance mutations and ALK amplification; and (2) ALK-independent resistance mechanisms, such as bypass signaling pathway activation and lineage changes [53]. 

Secondary ALK inhibitor resistance mutations were previously identified in 20–30% of tumor samples with crizotinib failure [54]. Among the crizotinib-resistant secondary mutations, L1196M and G1269A are the most frequently detected in clinical samples [55]. The L1196M and G1269A mutations are located in the ATP-binding pocket and hinder crizotinib binding. Other crizotinib-resistant mutations include I1151T, L1152P/R, C1156Y, G1128A, I1171T/N/S, F1174V, E1201K, G1202R, S1206C/Y, and V11180L [56,57,58], which may enhance the ATP-binding affinity and enzymatic activity of the kinase. 

Second-generation ALK inhibitors, such as ceritinib, alectinib, and brigatinib, have been developed and approved clinically to overcome crizotinib-resistant mutations. These inhibitors are potent against common crizotinib-resistant mutations, L1196M, and G1269A [32,59,60,61]. Patients whose ALK-p NSCLC conditions have been treated with second-generation ALK inhibitors inevitably develop acquired resistance, and secondary acquired resistance mutations have been determined in approximately 50–70% of these patients. Figure 2 shows the pharmacological activities of alectinib and lorlatinib against ALK fusion proteins with resistant mutations. The G1202R mutation occurs in the solvent-front ATP-binding site region of ALK and weakens the binding of all first- and second-generation ALK inhibitors because of steric hindrance [62,63]. 

Although the G1202R mutation is rarely detected in crizotinib-relapsed clinical samples (2%), it is the most frequent resistance mutation following the administration of second-generation ALK inhibitors, accounting for 40–65% of all acquired resistance mutations [31,64,65]. Second-generation ALK inhibitors have greater potency than crizotinib; thus, resistance mutation purification is suggested. Given its poor potency against ALK kinase activity, crizotinib may select less potent resistance mutations. Alectinib-resistant mutations include G1202R and I1171N. Interestingly, heterogeneous tumor evolution and high tumor mutation levels possibly contribute to the rapid acquisition of alectinib resistance [66]. Ceritinib resistance mutations include T1151K, T1151R, F1174V, and G1202R [58,67,68], whereas brigatinib resistance mutations include D1203N and E1210K [69,70].

In 2018 lorlatinib was approved by the U.S. FDA for the treatment of patients with ALK-p NSCLC. Lorlatinib displays activity against all potential ALK-TKI resistance mutations, including L1196M, G1269A, and G1202R [64]. Lorlatinib is considered to be one of ALK inhibitors that may hold promise for overcoming the high frequency of ALK inhibitor resistance mutations, particularly G1202R [64]. Next-generation sequencing results showed that ALK mutations in several lorlatinib-resistant individuals accumulate during consecutive dosing of an ALK inhibitor. ALK combined mutations reported to date include L1196M/D1203N, F1174L/G1202R, and C1156Y/G1269A [71]. ALK D1203N is more common during failure of lorlatinib than during failure of second-generation ALK inhibitors [69]. Interestingly, some ALK inhibitor-related compound mutations conferring lorlatinib resistance led to re-sensitization to first- or second-generation ALK inhibitors [13]. ALK amplification involves an ALK-dependent resistance mechanism. Crizotinib causes ALK gene amplification as a resistance mechanism; however, its occurrence is rarer than that of ALK resistance mutations. Moreover, second- and third-generation ALK inhibitors are not identified, suggesting that they may not be clinically relevant as highly potent ALK inhibitors [64]. 

Activating bypass signaling pathways is important in ALK-independent resistance mechanisms via gene alterations, autocrine signaling with ligand overexpression, and feedback signaling. Such pathways include epidermal growth factor receptor signaling [30,72], KIT amplification [31], MET amplification [73,74,75], IGF-1R activation [76], BRAF V600E mutation [74], and increased expression of the MET ligand of hepatocyte growth factor [77]. Different from the previous addicted ALK activation, the bypass signal activation leads to the activation of downstream factors such as the RAF/MEK/ERK and PI3K/AKT pathways, and provides survival signals. P-glycoproteins (P-gp) encoded by the multidrug resistance 1 (*MDR1*) gene can induce multidrug resistance through the ATP-dependent efflux of chemotherapeutical agents [78]. P-gp actively excludes the substrate from the blood at the BBB, thereby limiting CNS penetration [79]. In most patients with crizotinib failure the CNS is the primary metastatic site [80]. Brain accumulation of ceritinib is restricted by P-gp and BCRP [81]; otherwise, alectinib and lorlatinib are non P-gp substrates that can achieve higher concentrations in the CNS. As a potential resistance mechanism, P-gp overexpression was determined in patients with tumor tissues of crizotinib- and ceritinib-resistant ALK mutant NSCLC [82]. Phenotypic changes are also a mechanism of ALK inhibitor resistance in ALK-mutated NSCLC. Epithelial-to-mesenchymal transition (EMT) and small cell lung cancer (SCLC) or squamous cell carcinoma conversion have been reported after ALK inhibitor therapy for ALK-p adenocarcinoma [83,84,85,86]. Histological changes in tumors from adenocarcinoma to SCLC have been reported in 3–10% of patients with EGFR-TKI-resistant NSCLC [87]. The histological changes may be associated with retinoblastoma loss acquisition and genetic/epigenetic features of SCLC, such as EGFR-TKI resistance [88]. With regard EMT, molecular mechanisms such as inhibitor resistance are unknown. Moreover, histone deacetylase inhibitors can overcome this EMT-mediated ALK inhibitor resistance by reversing EMT pre-clinically.

### 1.5. Significance of the Present Meta-Analysis

Based on these prospects, we conducted a comprehensive literature search and network meta-analysis (NMA; UMIN 000049680). The results of this meta-analysis provide important information to guide clinical oncologists treating non-small cell lung cancer when considering treatment strategies for patients with ALK-p, ALK inhibitor-naive advanced NSCLC.

## 2. Results

### 2.1. Systematic Review

A systematic literature search identified 2724 studies (478 from PubMed [89], 834 from Cochrane Central Register of Controlled Trials [CENTRAL] [90], 356 from EMBASE [91], and 1056 from SCOPUS [92]), with 1907 that remained after removing duplicates. After employing the Patients, Interventions, Comparisons, Outcomes, and Study Design (PICOS) approach, nine research studies were selected for inclusion in the NMA, of which two articles compared crizotinib with platinum based chemotherapy (PROFILE1014 [28] and PROFILE1029 [29]), three compared alectinib with crizotinib (ALEX [35], J-ALEX [37], and ALESIA [38]), and one each compared ceritinib with chemotherapy, brigatinib with crizotinib, lorlatinib with crizotinib, and ensartinib with crizotinib (ACEND-4 [36], ALTA-1L [34], CROWN [33], and eXalt3 [41], respectively). The study selection process is summarized in Figure 3. The primary inclusion criteria are summarized in Appendix A, and the main characteristics of the included studies are summarized in Appendix A. The data of 2484 patients from the nine studies (chemotherapy: 461, crizotinib: 1025, ceritinib: 189, alectinib: 380, brigatinib: 137, lorlatinib: 149, ensartinib 143) were used for the analysis. A network map of the present network meta-analysis is shown in Figure 4.

### 2.2. Comparison of ALK Inhibitors by Generation

#### 2.2.1. PFS in Overall Patients

The efficacies of chemotherapy, first-generation ALK inhibitors (crizotinib), second-generation ALK inhibitors (ceritinib, alectinib, brigatinib, and ensartinib), and third-generation ALK inhibitors (lorlatinib) in prolonging PFS were compared between generations in the overall patient population of ALK-p, ALK inhibitor-naive advanced NSCLC. Statistically significant differences were found among all generations compared (3rd vs. 2nd, 3rd vs. 1st, 2nd vs. 1st, 3rd vs. Chemo, 2nd vs. Chemo, and 1st vs. Chemo) (Figure 5a, Appendix A). Ranking by generation showed that third-generation ALK inhibitors had the best PFS benefit, followed by second-generation ALK inhibitors, first-generation ALK inhibitors, and chemotherapy (Appendix A). 

#### 2.2.2. PFS in a Subgroup of Patients with CNS Metastases

The efficacies of chemotherapy, first-generation ALK inhibitors (crizotinib), second-generation ALK inhibitors (ceritinib, alectinib, brigatinib, and ersatinib), and third-generation ALK inhibitor (lorlatinib) in prolonging the PFS in a subgroup of patients with CNS metastases were compared. Statistically significant differences were found between third- and second-generation ALK inhibitors, third- and first-generation ALK inhibitors, second- and first-generation ALK inhibitors, third-generation ALK inhibitors and chemotherapy, and second-generation ALK inhibitors and chemotherapy, but not between first-generation ALK inhibitors and chemotherapy (Figure 5b, Appendix A). Ranking by generation showed that third-generation ALK inhibitors had the highest PFS efficacy, followed by second-generation ALK inhibitors, first-generation ALK inhibitors, and chemotherapy (Appendix A). 

### 2.3. Comparison among ALK Inhibitors

#### 2.3.1. PFS in Overall Patients

A paired comparison of the efficacies of ensartinib, lorlatinib, brigatinib, alectinib, ceritinib, crizotinib, and chemotherapy in prolonging the PFS of the overall patients is presented in Appendix A. Ranking by drug showed that lorlatinib was the most effective in prolonging PFS, followed by alectinib, brigatinib, ensartinib, crizotinib, ceritinib, and chemotherapy (Appendix A).

#### 2.3.2. PFS in a Subgroup of Patients with CNS Metastases

A paired comparison of the efficacies of ensartinib, lorlatinib, brigatinib, alectinib, ceritinib, crizotinib, and chemotherapy in prolonging the PFS in the subgroup of patients with CNS metastases is presented in Appendix A. Ranking by drug showed that lorlatinib was the most effective in prolonging PFS, followed by brigatinib, alectinib, ensartinib, crizotinib, ceritinib, and chemotherapy (Appendix A). 

#### 2.3.3. PFS in Non-Asian Subgroup

A paired comparison of the efficacies of ensartinib, lorlatinib, brigatinib, alectinib, ceritinib, crizotinib, and chemotherapy in prolonging the PFS in non-Asian subgroup was presented in Appendix A. Ranking by drug showed that lorlatinib was the most effective in prolonging PFS, followed by alectinib, brigatinib, ensartinib, ceritinib, crizotinib, and chemotherapy (Appendix A).

#### 2.3.4. PFS in Asian Subgroup

A paired comparison of the efficacies of ensartinib, lorlatinib, brigatinib, alectinib, ceritinib, crizotinib, and chemotherapy in prolonging the PFS in the subgroup of patients with CNS metastases is presented in Appendix A. Ranking by drug showed that ensartinib was the most effective in prolonging PFS, followed by alectinib, brigatinib, lorlatinib, crizotinib, ceritinib, and chemotherapy (Appendix A).

### 2.4. Evaluation of Bias

The qualities of the studies that were included were appraised on the basis of the Cochrane-recommended Risk of Bias tool 2 (RoB2) [93]. Nine studies that were included in the present systematic review and NMA were judged as “some concerns” in the overall assessment. Specifically, they were all open-label studies judged as some concerns in terms of bias due to deviations from intended interventions or bias in measurement of the outcome. PROFILE 1029 [29] was also judged as some concerns in terms of bias arising from randomization because this process was not sufficiently detailed. No domains were identified as high risk (Appendix A).

### 2.5. Sensitivity Analysis

Of the nine studies included in this analysis, three (ALTA-1L [34], J-ALEX [37] and eXalt3 [41]) included a group of patients who had received partial chemotherapy. To address this heterogeneity, a sensitivity analysis [94,95] was performed by eliminating patients with previous exposure to chemotherapy from these three trials. Consequently, the paired comparison results of the four treatment groups were sustained (Appendix A). Further, comparable results were achieved for the ranking of the four treatment groups (Appendix A). These results indicated that the inclusion or exclusion of the patients with previous chemotherapy did not impact the overall definitive conclusions.

### 2.6. Assessment of Study-to-Study Heterogeneity

In addition, ALEX [35], J-ALEX [37], ALESIA [38], ALTA-L1 [34], and eXalt3 [41] compared crizotinib with second-generation ALK inhibitors. Therefore, we also evaluated the inter-trial heterogeneity of PFS in these five trials [96]. Results showed that I^2^ was 49.8% (*p* = 0.093), indicating mild between-trial heterogeneity (Appendix A).

## 3. Discussion

This review provides a comprehensive overview of future therapeutic strategies for ALK-p, ALK inhibitor-naïve advanced NSCLC with CNS metastases, mechanisms underlying acquired resistance, and strategies to overcome this challenge.

The efficacies of six ALK inhibitors (ensartinib, lorlatinib, brigatinib, alectinib, ceritinib, and crizotinib) were compared with that of chemotherapy in the overall patients with ALK-p, ALK inhibitor-naïve advanced NSCLC and in the subgroup of patients with CNS metastases. The comparisons were conducted by ALK inhibitor generation and by drug. Generation-specific comparison results showed that third-generation ALK inhibitors had the best efficacy in prolonging the PFS of the overall patient population and in the subgroup of patients with CNS metastases. Statistically significant differences in efficacy were found between third-generation ALK inhibitors and second- and first-generation ALK inhibitors in the overall patients and in the subgroup of patients with CNS metastases. Significant differences in efficacy in prolonging PFS were also demonstrated between second- and first-generation ALK inhibitors. Drug-specific comparison results showed that lorlatinib had the best efficacy in prolonging PFS in the overall patients, followed by alectinib, brigatinib, ensartinib, ceritinib, crizotinib, and chemotherapy. The differences between lorlatinib and brigatinib, ensartinib, ceritinib, crizotinib, and chemotherapy were statistically significant. Analysis of the subgroup of patients with CNS metastases showed that lorlatinib exerted the most favorable effect, followed by brigatinib, ensartinib, ceritinib, crizotinib, and chemotherapy. The differences between lorlatinib and ensartinib, ceritinib, crizotinib, and chemotherapy were statistically significant. 

Several previous meta-analyses have compared the efficacies of ALK inhibitors in patients with ALK-p, ALK inhibitor-naive advanced NSCLC [39,40,97,98,99,100,101,102,103,104]. However, a generation-specific comparison of the efficacies of six ALK inhibitors (ensartinib, lorlatinib, alectinib, brigatinib, ceritinib, and crizotinib) in patients with CNS metastases remains lacking. For the first time, we compared the efficiencies of these six ALK inhibitors by generation in prolonging the PFS of the overall patient population and a subgroup of patients with CNS metastases. Results showed that the third-generation ALK inhibitors were better than the other generations in prolonging the PFS of the overall patients and the subgroup of patients with CNS metastases, respectively. 

Notable findings were also obtained in drug-specific comparisons. For instance, lorlatinib was significantly better than brigatinib in prolonging the PFS of the overall patients, but their difference was not significant in the subgroup of patients with CNS metastases. In addition, evaluation results showed that brigatinib ranked third in the overall participant population but second, above alectinib, in the subgroup of patients with CNS metastases. These results support the theory that brigatinib, along with lorlatinib, is a potential first-line treatment option for ALK-p, ALK inhibitor-naïve advanced NSCLC with CNS metastases.

Our results also suggest that lorlatinib has potential as a novel first-line treatment for ALK-p, ALK inhibitor-naive advanced NSCLC. However, lorlatinib should not be recommended for all patients with this disease because its tolerability is reportedly lower than that of alectinib, and its effect on OS was not evaluated. Furthermore, in our analysis of racial differences lorlatinib ranked highest in PFS among non-Asians, whereas ensartinib ranked highest among Asians. Further clinical studies are warranted to develop a detailed treatment strategy for first-line treatment of ALK-p ALK-untreated advanced NSCLC.

This NMA has several limitations. First, the study compared the efficacies of six ALK inhibitors in the overall patients and subgroup of patients with CNS metastases. However, OS and safety outcomes were not analyzed because of insufficient data reported for CNS metastases. Further validation is needed to determine whether the results of this comparative analysis of PFS in the subgroup with CNS metastases will be consistent with the results of the comparative analyses of OS and safety outcomes. Second, this analysis included patients who had received systemic anticancer chemotherapy and those who had not. Although the results of sensitivity analysis showed that the inclusion and exclusion of patients who had undergone systemic anti-cancer chemotherapy did not apparently influence the results, we cannot completely rule out the potential impact of this heterogeneity on the final conclusions. Third, mild heterogeneity, although not statistically significant, was demonstrated in the five studies comparing second-generation ALK inhibitors with crizotinib (ALEX [35], J-ALEX [37], ALESIA [38], ALTA-L1 [34], and eXalt3 [41]). Although the NMA used a Bayesian model that assumed potential heterogeneity among the included studies, we cannot completely rule out the possibility that individual potential heterogeneity may have influenced the final conclusions. Finally, the number of included studies is as few as nine references, and we cannot completely exclude the possibility that the insufficient number of included studies may affect the convergence status of the models in the Bayesian network meta-analysis. To address this issue, the convergence status of our model was visually assessed. The results confirmed the favorable convergence status of our analysis. These results suggest that the number of studies covered was sufficient, at least in terms of model convergence.

## 4. Materials and Methods

### 4.1. Comprehensive Literature Search

A comprehensive literature search was conducted to identify relevant reports published from 1946 to the present. On 3 December 2022, four databases (PubMed [89], CENTRAL [90], EMBASE [91], and SCOPUS [92]) were searched for studies on NSCLC and ALK inhibitors by using keywords such as “ensartinib”, “lorlatinib”, “brigatinib”, “alectinib”, “ceritinib”, “crizotinib”, and their Medical Subject Headings terms. Appendix B shows the keywords used in the search. The strategy was also used for searching the EMBASE, CENTRAL, and SCOPUS databases to ensure comprehensiveness, robustness, and certainty of the search. The strategy used for searching PubMed was also used for searching EMBASE, CENTRAL, and SCOPUS. If data necessary for the analysis were not available from the journals, the authors were consulted by e-mailing the corresponding authors. The main purpose of this systematic review was to verify all publicized phase III clinical trials in order to make comparisons and rank the efficacy of the seven therapeutic groups in terms of efficacy; namely, ensartinib, lorlatinib, brigatinib, alectinib, ceritinib, crizotinib, and chemotherapy in patients with ALK-p advanced NSCLC. The analyses in this review were based on the Preferred Reporting Items for Systematic Review and Meta-analysis (PRISMA) guidelines [105] and the PRISMA extension statement for reporting of systematic reviews incorporating network meta-analyses of health care interventions (PRISMA-NMA) [106]. Two investigators (KA and AA) independently conducted the literature search. Inclusion and exclusion criteria were adapted to the retrieved studies using the PICOS approach to ensure the currency of indirect comparative analyses by handling potential heterogeneity in clinical and methodological aspects between studies. 

### 4.2. Quality Assessment

We assessed the quality of the RCTs included in the NMA using the RoB2 recommended by the Cochrane Collaboration [93]. The following parameters were rated as low risk, some concerns, or high risk: (1) bias arising from the randomization process; (2) bias due to deviations from the intended intervention; (3) bias due to missing outcome data; (4) bias in measurement of the outcome; and (5) bias in selection of the reported result. Evaluations were performed independently by two researchers (KA and SK), and any discrepancies were resolved by a third researcher (TY).

### 4.3. Inclusion Criteria (Pre-Defined PICOS)

#### 4.3.1. Patients

The following inclusion criteria were used: (1) age 18 years or older; (2) histologically or cytologically confirmed progressive or metastatic ALK-p NSCLC; (3) performance status of 0 to 2 (on a 5-point scale, higher numbers indicating more severe disability); (4) at least one measurable lesion assessed according to RECIST version 1.1,25; and (5) no prior exposure to ALK-targeted therapy.

#### 4.3.2. Intervention

In this analysis, patients treated with ensartinib (225 mg/day), lorlatinib (100 mg/day), brigatinib (180 mg/day), alectinib (300 or 600 mg/day), ceritinib (750 mg/day), crizotinib (250 mg/day), and platinum-based chemotherapy (all doses and dosage forms were approved, recommended, or specified in the Phase III study) were considered. Phase III trials that included any of these agents were eligible for inclusion. Crizotinib was the first approved ALK inhibitor and the former first-line agent for initial therapy, and platinum-based chemotherapy was the first-line agent for ALK-p treatment-naive NSCLC prior to the approval of crizotinib. Thus, crizotinib or platinum-based chemotherapy was assumed as the common comparator for each treatment. 

#### 4.3.3. Outcome

The primary efficacy endpoint was PFS in all participants and in the subgroup of patients with CNS metastases, with corresponding HRs and 95% credible intervals (CrIs). To rank the relative efficacy of each therapeutic approach, the surface under the cumulative rank under the curve (SUCRA) values were calculated for each endpoint, with higher SUCRA values corresponding to a more preferred therapeutic approach for the corresponding endpoints [107]. These analyses were performed on the overall participants and on each subgroup with CNMS. In addition, analyses were performed also by race (Asian and non-Asian) to account for demographics. To be eligible for this systematic review and NMA, the trial under analysis had to include at least one defined efficacy endpoint. These defined endpoints were analyzed only if data were available from the included trials. Two authors (KA and SK) independently extracted relevant data and resolved discrepancies in consultation with the third author (TY).

#### 4.3.4. Study Design

The research for this systematic review and meta-analysis was a phase III trial of a parallel-group RCT.

### 4.4. Statistical Analysis

The Bayesian NMA was performed following robustly established methods developed at the National Institute of Medical Research [108,109]. We applied a non-informative prior distribution, employed the standard Bayesian model described by Dias et al. [108,109], and assumed inconsistency and heterogeneity among the included studies. Gibbs sampling on the basis of a Markov chain Monte Carlo method was utilized to evaluate the posterior distribution of the effect size [110,111]. The number of iterations was set to 50,000, with the first 10,000 being a burn-in sample to eliminate the influence of initial values. Effect sizes were expressed as HR and its 95% CrI, and the difference in effect size between treatment groups for each endpoint was considered significant if the 95% CrI did not include 1. SUCRA values ranged from 0% to 100%, with higher SUCRA values indicating better treatment outcomes [107]. The Brooks–Gelman–Rubin (BGR) diagnostic method was also used for the convergent diagnosis of all comparisons [112,113]. Both visual and BGR diagnostics confirmed the convergence of the model. OpenBUGS 1.4.0 (MRC Biostatistics Unit, Cambridge Public Health Research Institute, /jk, Cambridge, UK) was used for the Bayesian analysis, and STATA (ver. 14, StataCorp., College Station, TX, USA) was used to visualize the results (College Station, TX, USA). 

### 4.5. Sensitivity Analysis

A sensitivity analysis [95] was conducted by including or excluding research that was deemed heterogeneous based on the existence of conceptual heterogeneity between the included studies. This analysis was performed to evaluate whether the inclusion or exclusion of conceptually heterogeneous studies impacts the overall final conclusions.

### 4.6. Assessment for between-Study Heterogeneity

We evaluated the statistical heterogeneity among the included studies to determine whether it impacts the final conclusions [96]. Statistical heterogeneity between studies was expressed as I2 statistic (%). A heterogeneity between 30% and less than 50% was considered to indicate mild heterogeneity between studies, between 50% and 70% moderate heterogeneity, and greater than 70% high heterogeneity. The I2 statistic was calculated using pairwise meta-analysis with a random-effects model.

### 4.7. Ethical Aspects

Institutional Review Board approval and patient consent were waived because of the retrospective nature of this systematic review.

## 5. Conclusions

This review outlines future treatment strategies and future prospects for ALK-p, ALK inhibitor-naïve advanced NSCLC with CNS metastasis, with a focus on elucidating and overcoming acquired resistance mechanisms. In addition, the therapeutic efficacies of ALK inhibitors in prolonging the PFS of the overall patients with ALK-p, ALK inhibitor-naïve advanced NSCLC and a subgroup of patients with CNS metastases are compared by drug and by generation. Generation-specific comparison shows that third-generation ALK inhibitors are significantly more efficient than second-generation and first-generation ALK inhibitors in prolonging the PFS of the overall patients and subgroup of patients with CNS metastases. Drug-specific comparison demonstrates that lorlatinib is the most efficient in prolonging the PFS of the overall patients and subgroup of patients with CNS metastases. Notably, although a significant difference in efficacy of prolonging PFS was found between lorlatinib and brigatinib in the overall patient population, no such significant difference was found in the subgroup of patients with CNS metastases. These results indicate a trend toward brigatinib as a promising first-line treatment option along with lorlatinib in the subgroup of patients with CNS metastases. These results can serve as a foundation for basic, clinical, and translational research and guide clinical oncologists in developing individualized treatment strategies for ALK-p ALK inhibitor-naïve advanced NSCLC. This NMA includes direct and indirect comparisons, and additional studies are warranted to confirm the results. The results of this analyses can serve as a basis for further clinical studies formulating novel treatment strategies for ALK-p ALK inhibitor-naïve advanced NSCLC with CNS metastases or acquired resistance mutations.

## Figures and Tables

**Figure 1 ijms-24-02242-f001:**
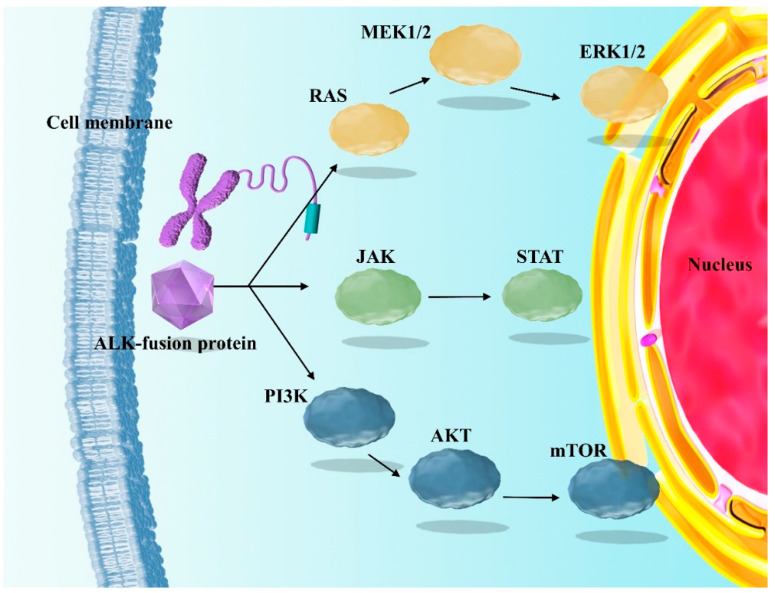
Mechanisms of ALK translocation cancer progression. EML4-ALK translation triggers the initiation of the PI3K-AKT, RAS, and JAK/STAT signaling cascades that influence tumor growth, proliferation, and viability. ERK, extracellular signal-regulated kinase EML4-ALK, echinoderm microtubule-associated protein-like 4-anaplastic lymphoma kinase; PI3K, phosphatidylinositol-3 kinase; mTOR, STAT, signal transducer and activator of transcription. ALK, anaplastic lymphoma kinase; mammalian target of rapamycin; JAK, Janus kinase; MEK, mitogen-activated extracellular signal regulated kinase; RAS, reticular activating system.

**Figure 2 ijms-24-02242-f002:**
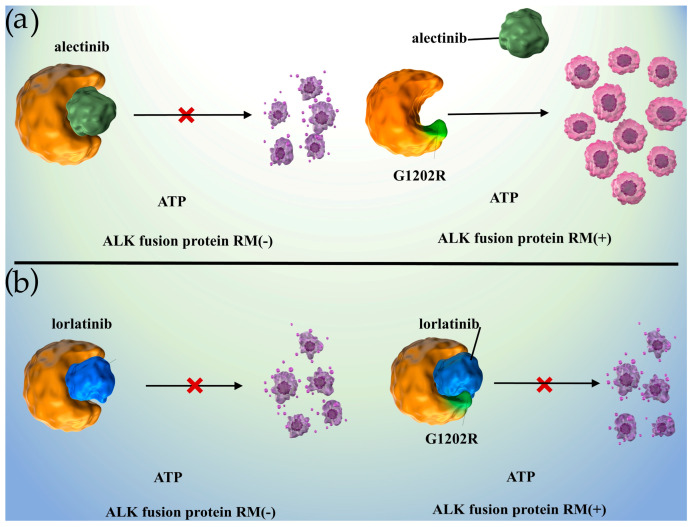
Different pharmacological activities of (**a**) alectinib and (**b**) lorlatinib toward ALK fusion proteins with resistance mutations. (**a**) Resistance mutations (e.g., G1202R) prevent alectinib from combining with the ATP-binding domain of the ALK fusion protein. (**b**) Lorlatinib successfully associated with the ALK fusion protein’s ATP-binding pocket with resistance mutations, and downstream signals associated with tumor progression are downregulated; ATP, adenosine triphosphate; RM, resistance mutation; ALK, anaplastic lymphoma kinase.

**Figure 3 ijms-24-02242-f003:**
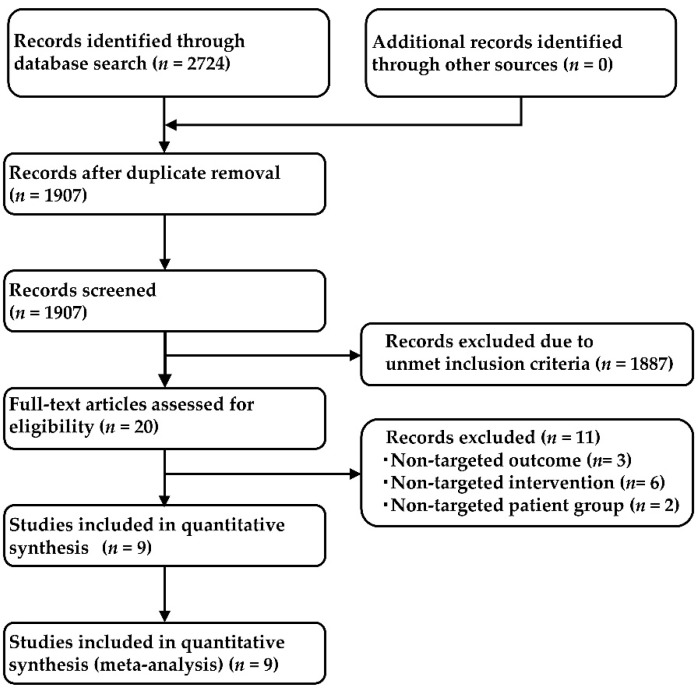
Process diagram of the research selection.

**Figure 4 ijms-24-02242-f004:**
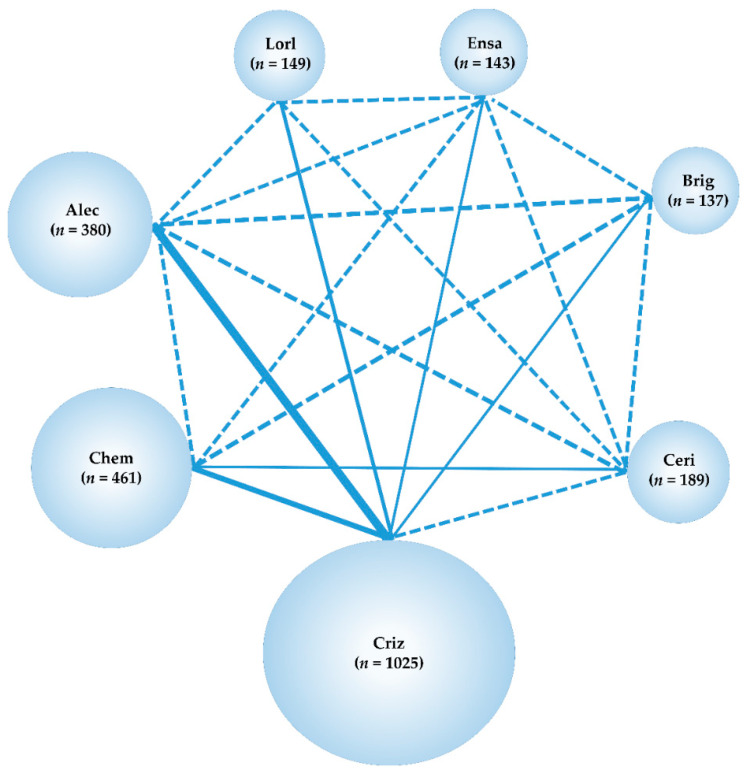
Network map of the seven therapeutic groups: ensartinib, lorlatinib, brigatinib, alectinib, ceritinib, crizotinib, and chemotherapy. In this network map randomized controlled trials (RCTs) were represented by a solid line, with the breadth of the solid line correlated with the numbers of studies included. Broken lines represent no head-to-head RCTs and trial to comparison of treatments. *n* is the total number of patients in each group; Ensa, ensartinib; Lorl, lorlatinib; Brig, brigatinib; Alec, alectinib; Criz, crizotinib; Ceri, ceritinib; Chem, chemotherapy.

**Figure 5 ijms-24-02242-f005:**
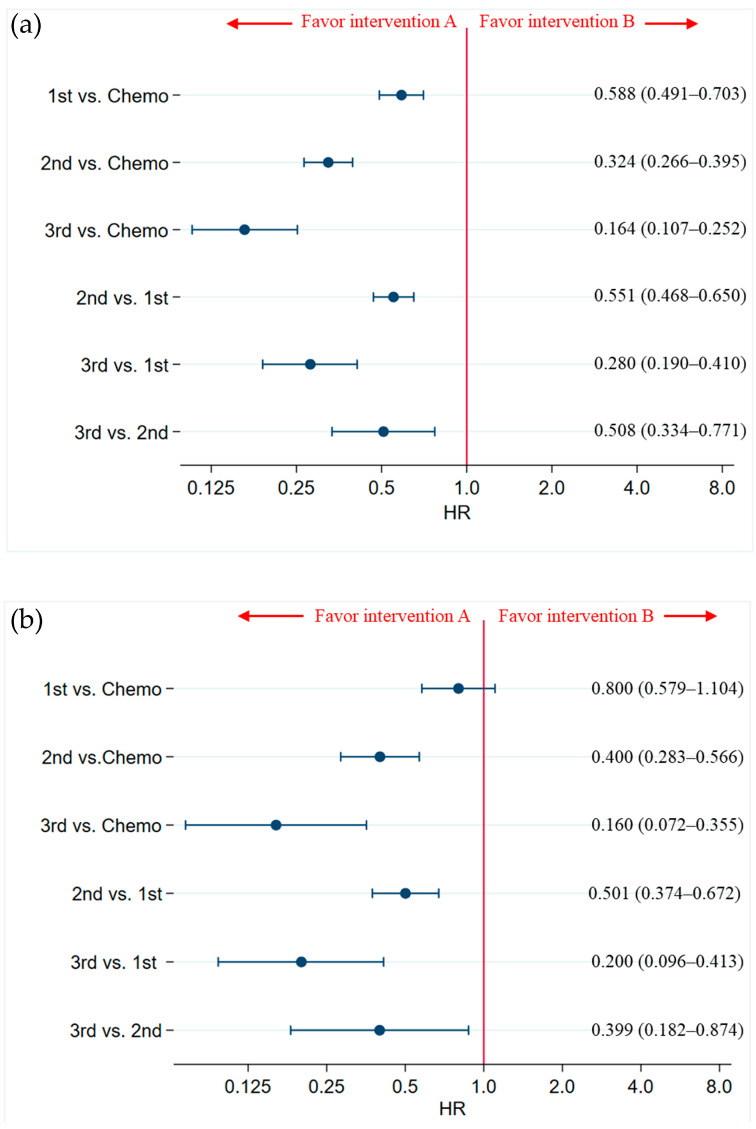
Comparison of the efficacies of chemotherapy, first-generation ALK inhibitors (crizotinib), second-generation ALK inhibitors (ceritinib, alectinib, brigatinib, and ensartinib), and third-generation ALK inhibitors (lorlatinib) in prolonging the PFS of (**a**) overall patients with ALK-p, ALK inhibitor-naive advanced NSCLC and (**b**) a subgroup of patients with CNS metastases. Data are expressed as hazard ratios (HRs) and 95% credible intervals (CrIs); ALK, anaplastic lymphoma kinase; ALK-p, anaplastic lymphoma kinase rearrangement positive; NSCLC, non-small cell lung cancer; CNS, central nervous system.

## Data Availability

The authors affirm that the analyzed datasets in this present work are available from the corresponding author upon reasonable request.

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
