# Peer review of "Comparative Efficacy of ALK Inhibitors for Treatment-Naïve ALK-Positive Advanced Non-Small Cell Lung Cancer with Central Nervous System Metastasis: A Network Meta-Analysis"

_ijms, 2023, doi:10.3390/ijms24032242_

Round 1

Reviewer 1 Report

The authors need to define if it is a review or meta-analysis clearly. The manuscript needs major restructuring.

Also, the authors should justify the significance of the study. As for now, the study merely sounds like a discussion of literature. Hence, it may not be sufficient to be published in the Q1 journals with good Impact Factor as International Journal of Molecular Sciences.

Also, references are missing in some places, eg. Introduction (para 4-5).

The authors need to justify if 9 studies are sufficient to draw any strong conclusion. Also, were the patient demographics taken into consideration in the evaluation? 

Author Response

Dear Reviewer 1

Thank you for giving us the opportunity to resubmit to International Journal of Molecular Sciences a revised draft of our manuscript titled “Comparative Efficacy of ALK Inhibitors for Treatment-Naïve ALK-Positive Advanced Non-Small Cell Lung Cancer with Central Nervous System Metastasis: A Network Meta-Analysis” (Manuscript ID: ijms-2148951) (The title has been changed to address the reviewer's suggestion).

We sincerely appreciate the time and effort that you and the reviewers have dedicated to providing your valuable feedback on our manuscript. We are grateful to the reviewers for their insightful comments, and have been able to incorporate changes to reflect most of their suggestions. We have highlighted in yellow the changes within the manuscript.

Here is a response to the reviewers’ comments.

Comment1: The authors need to define if it is a review or meta-analysis clearly. The manuscript needs major restructuring.

Response1: We sincerely appreciate your valuable suggestions. We concur with the reviewer's comments. We clearly declare that this paper is a meta-analysis. To represent this more clearly, we have changed the title to indicate that this is a report of a meta-analysis (network meta-analysis). We have made major changes to the overall framework of the paper. That is, the parts of the manuscript that corresponded to Sections 2, 3, and 4 before the revision were incorporated into the Introduction section, and the entire revised manuscript now consists of the Introduction, Results, Discussion, Methods, and Conclusion. In addition, A new part, “1.5. Significance of the present meta-analysis”, was established at the end of the introduction section (L307–312). Further, the significance of the study was again clearly stated and emphasized at the end of the introduction section as follows;” Based on these prospects, we conducted a comprehensive literature search and network meta-analysis (NMA; UMIN 000049680). The results of this meta-analysis provide important information to guide clinical oncologists treating non-small cell lung cancer when considering treatment strategies for patients with ALK-p, ALK inhibitor-naive advanced NSCLC”(L308–312).

Comment2: Also, the authors should justify the significance of the study. As for now, the study merely sounds like a discussion of literature. Hence, it may not be sufficient to be published in the Q1 journals with good Impact Factor as International Journal of Molecular Sciences.

Response2: Thank you very much for your very accurate remarks. We agree with the reviewer's comments. We have reiterated the significance of this study by stating the following at the end of the Introduction section in revised manuscript as follows;” Based on these prospects, we conducted a comprehensive literature search and net-work meta-analysis (NMA; UMIN 000049680). The results of this meta-analysis provide important information to guide clinical oncologists treating non-small cell lung cancer when considering treatment strategies for patients with ALK-p, ALK inhibitor-naive advanced NSCLC”(L308–312).

Comment3: Also, references are missing in some places, eg. Introduction (para 4-5).

Response3: Thank you very much for pointing out this very important point. We agree with the reviewer's comments. To address this issue, we have cited additional relevant references in several places, including paragraphs 4-5(L84, L90, and L181).

Comment4: The authors need to justify if 9 studies are sufficient to draw any strong conclusion.

Response4: I sincerely appreciate your very valuable and important comments. We agree with the reviewer's comments. We cannot completely exclude the possibility that the insufficient number of included studies may affect the convergence status of the models in the Bayesian network meta-analysis. We have validated the convergence status of our model to address this issue. As a result, the favorable convergence status of the present analysis was confirmed. These results suggest that the number of included studies was sufficient in terms of model convergence. In this regard, we have added the following to the limitation part of the discussion section; “The number of included studies is as few as 9 references, and we cannot completely exclude the possibility that the insufficient number of included studies may affect the convergence status of the models in the Bayesian network meta-analysis. To address this issue, the convergence status of our model was visually assessed. The results confirmed the favorable convergence status of our analysis. These results suggest that the number of studies covered was sufficient at least in terms of model convergence” (L484–490).

Comment5: Also, were the patient demographics taken into consideration in the evaluation? 

Response5: We sincerely appreciate your raising this very important point. We agree with the reviewer's comments. We performed an additional analysis by drug by race to consider patient demographics. The results were summarized in result section, and detailed in Tables S7 and S8, which were newly added. (With this change, Tables S7 and S8 in the pre-revision manuscript have been renamed Tables S9 and S10, respectively). Further, we have added a description of the results in the discussion section as follows; “Furthermore, in our analysis of racial differences, lorlatinib ranked highest in PFS among non-asians, whereas ensartinib ranked highest among asians”(L465–L467).

We are confident that our revised manuscript will be suitable for publication in International Journal of Molecular Sciences and look forward to receiving your editorial decision.

Thank you for your consideration.

Sincerely,

Koichi Ando

Division of Respirology and Allergology, Department of Medicine, Showa University School of Medicine

1-5-8 Hatanodai, Shinagawa-ku, Tokyo, 142-8666, Japan

Tel: +81-3-3784-8532

Fax: +81-3-3784-8742

Reviewer 2 Report

The authors conducted an interesting study of TKIs with an important role in the treatment of ALK-p lung cancer, using a network meta analysis approach. The method also makes sense.

One point that concerns me is the second paragraph on p. 7, where the authors state that , lorlatinib is the only ALK inhibitor that can overcome the high frequency of ALK inhibitor resistance mutations, However, in the report by mizuta et al. (1) brigatinib may also overcome G1202R, so I think this point is still open for discussion.

(1)Mizuta, H., Okada, K., Araki, M. et al. Gilteritinib overcomes lorlatinib resistance in ALK-rearranged cancer. Nat Commun 12, 1261 (2021). 

Author Response

Dear Reviewer 2

Thank you for giving us the opportunity to resubmit to International Journal of Molecular Sciences a revised draft of our manuscript titled “Comparative Efficacy of ALK Inhibitors for Treatment-Naïve ALK-Positive Advanced Non-Small Cell Lung Cancer with Central Nervous System Metastasis: A Network Meta-Analysis” (Manuscript ID: ijms-2148951) (The title has been changed to address the reviewer's suggestion).

We sincerely appreciate the time and effort that you and the reviewers have dedicated to providing your valuable feedback on our manuscript. We are grateful to the reviewers for their insightful comments, and have been able to incorporate changes to reflect most of their suggestions. We have highlighted in yellow the changes within the manuscript.

Here is a response to the reviewers’ comments.

Comment1: The authors conducted an interesting study of TKIs with an important role in the treatment of ALK-p lung cancer, using a network meta-analysis approach. The method also makes sense. One point that concerns me is the second paragraph on p. 7, where the authors state that, lorlatinib is the only ALK inhibitor that can overcome the high frequency of ALK inhibitor resistance mutations, However, in the report by Mizuta et al. (1) brigatinib may also overcome G1202R, so I think this point is still open for discussion.

  • Mizuta, H., Okada, K., Araki, M. et al. Gilteritinib overcomes lorlatinib resistance in ALK-rearranged cancer. Nat Commun 12, 1261 (2021).

Response1: Thank you sincerely for your very valuable and important points. We concur with the reviewer's comments. We have modified the description of the relevant section as follows; ” Lorlatinib is considered to be one of ALK inhibitors that may hold promise for overcoming the high frequency of ALK inhibitor resistance mutations, particularly G1202R”(L267–269).

We are confident that our revised manuscript will be suitable for publication in International Journal of Molecular Sciences s and look forward to receiving your editorial decision.

Thank you for your consideration.

Sincerely,

Koichi Ando

Division of Respirology and Allergology, Department of Medicine, Showa University School of Medicine

1-5-8 Hatanodai, Shinagawa-ku, Tokyo, 142-8666, Japan

Tel: +81-3-3784-8532

Fax: +81-3-3784-8742

Round 2

Reviewer 1 Report

Thank you to the authors for their detailed response. I recommend accepting the manuscript.